

# Drift macroalgal distribution in northern Gulf of Mexico seagrass meadows

Kelly M. Correia[1,2], Scott B. Alford[3], Benjamin A. Belgrad[1], Kelly M. Darnell[4], M. Zachary Darnell[4], Bradley T. Furman[5], Margaret O. Hall[5], Christian T. Hayes[4,6], Charles W. Martin[3], Ashley M. McDonald[3] and Delbert L. Smee[1,2]

[1] Dauphin Island Sea Lab, Dauphin Island, AL, United States of America
[2] University of South Alabama, Mobile, AL, United States of America
[3] Nature Coast Biological Station, University of Florida, Cedar Key, FL, United States of America
[4] Division of Coastal Sciences, School of Ocean Science and Engineering, The University of Southern Mississippi, Ocean Springs, MS, United States of America
[5] Florida Fish and Wildlife Conservation Commission, Fish and Wildlife Research Institute, St. Petersburg, FL, United States of America
[6] Department of Biology, Environmental Science, and Health Science, Waynesburg University, Waynesburg, PA, United States of America

Corresponding author
Kelly M. Correia,
drkmcorreia@gmail.com

## ABSTRACT

Drift macroalgae, often found in clumps or mats adjacent to or within seagrass beds, can increase the value of seagrass beds as habitat for nekton via added food resources and structural complexity. But, as algal biomass increases, it can also decrease light availability, inhibit faunal movements, smother benthic communities, and contribute to hypoxia, all of which can reduce nekton abundance. We quantified the abundance and distribution of drift macroalgae within seagrass meadows dominated by turtle grass *Thalassia testudinum* across the northern Gulf of Mexico and compared seagrass characteristics to macroalgal biomass and distribution. Drift macroalgae were most abundant in areas with higher seagrass shoot densities and intermediate canopy heights. We did not find significant relationships between algal biomass and point measures of salinity, temperature, or depth. The macroalgal genera *Laurencia* and *Gracilaria* were present across the study region, *Agardhiella* and *Digenia* were collected in the western Gulf of Mexico, and *Acanthophora* was collected in the eastern Gulf of Mexico. Our survey revealed drift algae to be abundant and widespread throughout seagrass meadows in the northern Gulf of Mexico, which likely influences the habitat value of seagrass ecosystems.

## INTRODUCTION

Drift macroalgae often originate as attached algae on seagrass leaves and other hard substratum before becoming uprooted by various physical disturbances (*e.g.*, currents, waves) (*Norton & Mathieson, 1983*; *Bell & Hall, 1997*; *Biber, 2002*; *Lirman et al., 2003*). They are commonly found in small patches within and around the calm, coastal seagrass meadows from early spring to mid-summer (*Norton & Mathieson, 1983*), with distributions
influenced by water currents as well as the roughness of the surrounding substrate (*Bell, Hall & Robbins, 1995*; *Bell & Hall, 1997*; *Biber, 2007*; *Fonseca & Koehl, 2006*). The ecosystem benefits of seagrass habitats are strongly correlated with their structural complexity, and macroalgal communities often further increase this complexity 3-100-fold (*Morris & Hall, 2001*; *Kingsford, 1995*). Seagrass and macroalgae are major constituents in some of the most productive coastal ecosystems and they enhance fisheries by providing valuable nursery habitat for a variety of finfish and invertebrate fauna (*e.g.*, *Carr, 1991*; *Jackson et al., 2001*; *Heck Jr, Hays & Orth, 2003*; *Guido et al., 2004*; *Bos et al., 2007*). In the Mediterranean, seagrass and macroalgal habitats support 30–40% of the commercial fish and 29% of recreational fish during their juvenile life stages (*Jackson, Wilding & Attrill, 2015*). The structure provided by seagrass and macroalgal communities can enhance feeding and growth rates, while lowering predation rates for many shrimp, crab, and fish species (*Orth, Heck Jr & Van Montfrans, 1984*; *Kingsford & Choat, 1985*; *Bax, 1998*; *Rooker, Holt & Holt, 1998*; *Nagelkerken et al., 2002*). The detached nature of drift macroalgal communities can also aid in dispersal of many small fish and invertebrates utilizing this structure (*Astill & Lavery, 2001*; *Holmquist, 1994*).

While macroalgae can enhance the ecosystem services of seagrasses, at sufficiently high biomass, macroalgae can lead to declines in organismal abundance and species diversity (*e.g.*, *Hull, 1987*; *Bonsdorff, 1992*; *Zajac & McCarthy, 2015*). The degree of this change may be tied to species-specific morphological traits (*Barthol Omew, Diaz & Cicchetti, 2000*). Green filamentous macroalgae in the Baltic Sea, for example, is problematic at high concentrations, causing hypoxia and altering the resident benthic communities (*Vahteri et al., 2000*). Massive influxes of the brown alga *Sargassum* to coastal systems have also led to similar declines in flora and fauna in the Caribbean, causing benthic mortality and decreasing habitat value (*Chávez et al., 2020*). However, brown algae in New Zealand led to an increase in fish and invertebrate abundance, with higher species abundances relative to attached vegetation and open water areas (*Kingsford & Choat, 1985*). Understanding the algal composition throughout the northern Gulf of Mexico and seasonal changes in biomass may allow us to better understand the species-specific effects that algae have on seagrass and their associated nekton communities.

Algae are classified into three evolutionarily distinct lineages based on variations in their morphological characteristics and tissue pigment composition, and consist of brown algae (Phaeophyceae), green algae (Chlorophyta), and red algae (Rhodophyta), with rhodophytes being the dominant taxa in the northern Gulf of Mexico (*Virnstein & Carbonara, 1985*; *Holmquist, 1997*). Over the last few decades, seagrass areal coverage has declined in many areas, including the northern Gulf of Mexico and southern Atlantic estuaries of North America (*Hall et al., 1999*; *Peneva, Griffith & Carter, 2008*; *Carter et al., 2011*). Macroalgal blooms have increased in frequency and intensity (*e.g.*, *Benz, Eiseman & Gallaher, 1979*; *Virnstein & Carbonara, 1985*; *Zieman, Fourqurean & Iverson, 1989*; *Kopecky & Dunton, 2006*; *Fredericq et al., 2009*) and are predicted to proliferate under future scenarios of warmer sea surface temperatures and ocean acidification (*Brodie et al., 2014*). Consequently, the functional role of drift algae may increase in importance in regions where seagrasses have declined. However, drift algae are known for being

highly variable across space and time (*Benz, Eiseman & Gallaher, 1979*; *Bell & Hall, 1997*), with their abundance and movement within estuaries rarely quantified and difficult to track, complicating our understanding of drift algae in seagrass ecosystems. Although underappreciated for their habitat value, drift macroalgae likely augment the value of seagrass beds as habitat for nekton by increasing the physical structure within and adjacent to seagrass beds. We surveyed seagrass meadows across the northern Gulf of Mexico and quantified (1) the abundance of drift macroalgae in early and late summer and (2) the relationship between algal abundance and environmental and seagrass metrics.

## MATERIALS & METHODS

### Study regions

Five estuaries in the northern Gulf of Mexico, each containing at least 20 sites within seagrass meadows, were surveyed twice during the early (May–June) and late (August–September) summer of 2018 (Fig. 1). Sites were selected by overlaying a tessellated hexagonal grid (500 m edge length) on each estuary in ArcGIS (*Moore, 2009*; *Neckles et al., 2012*; *Wilson & Dunton, 2012*). A randomly generated site within each of 20 to 25 grid cells that contained more than 50% seagrass coverage and a minimum of 500 m separation were selected for assessment (*Belgrad et al., 2021*). Across all regions, turtle grass (*Thalassia testudinum*) was the dominant macrophytic taxon; however, manatee grass (*Syringodium filiforme*) and shoal grass (*Halodule wrightii*) were also common. Star grass (*Halophila engelmannii*) and widgeon grass (*Ruppia maritima*) were present but occurred in <0.01% of surveys and are not considered in this study. *In situ* measurements of seagrass coverage ranged from 0–100%. Measurements were collected across twenty sites within Laguna Madre, TX (LM; 26°08′N, 97°14′W), Corpus Christi Bay, TX (CB; 27°51′N, 97°08′W), and the Chandeleur Islands, LA (LA; 29°54′N, 88°50′W). In Florida, measurements were collected from 25 sites at both Cedar Key (CK; 29°05′N, 83°01′W) and Charlotte Harbor (CH; 26°04′N, 82°14′W). At each sampling location, abiotic conditions (*i.e.*, temperature, salinity, DO, depth) were recorded, drift macroalgae biomass measured, and seagrass cover/abundance and morphometrics assessed. Because we sampled synoptically using the same methods, we were able to assess both algal biomass and distribution within and among locations across the northern Gulf of Mexico (Table 1; *Belgrad et al., 2021*; *Correia, 2021*).

### Drift algal abundance assessment

Within each hexagon, drift algal abundance within seagrass meadows was measured using a flat otter trawl, an epibenthic sled, and 1-m$^2$ quadrats. The use of these three sampling techniques provided valuable information about broad and fine scale macroalgal distributions within seagrass habitats. While the trawl covers a larger area than the epibenthic sled, it can quickly become fouled by high biomass of drift algae, making it difficult to standardize trawl lengths. Epibenthic sleds allow for a more standardized comparison between seagrass and macroalgal habitat within a given area. Each sled tow was pulled within seagrass habitats for the same distance allowing for a precise representation of macroalgae within each seagrass bed. Quadrats were used in the same vicinity as the epibenthic sled and trawl to assess seagrass and algal percent cover.
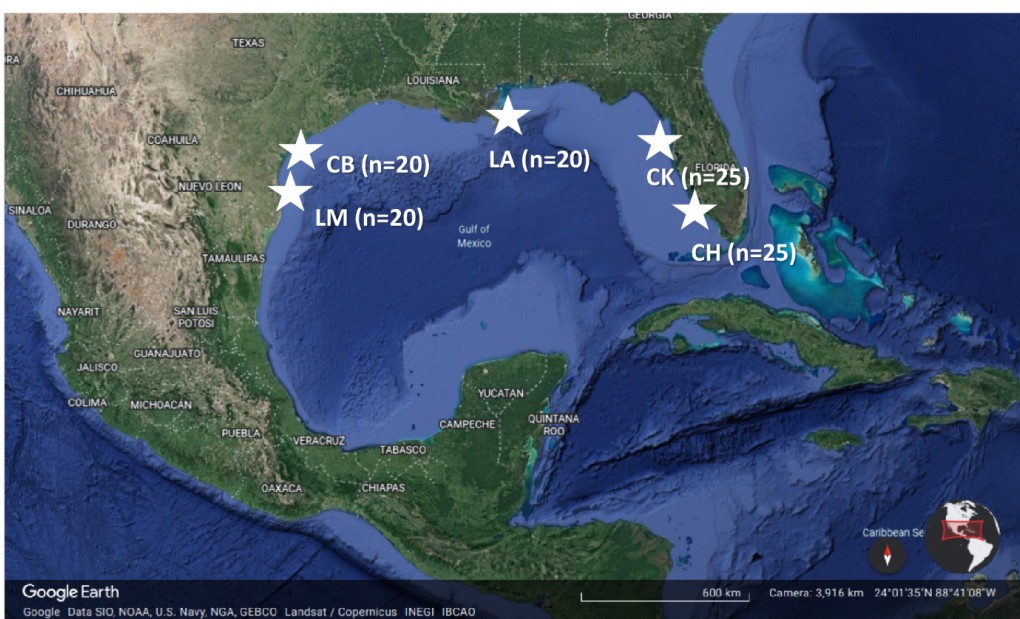

**Figure 1 Location of the five study estuaries (stars) throughout the northern Gulf of Mexico.** Regions include Laguna Madre, TX (LM), Corpus Christi Bay, TX (CB), the northern extent of the Chandeleur Islands (LA), Cedar Key, FL (CK), and Charlotte Harbor, FL (CH). n represents the number of sites that were sampled within each estuary during the early and late summer 2018. Map data ©2021 Google.

A 4.85-m flat otter trawl with a 3.8-cm stretch mesh body and 1.3-cm stretch mesh bag was towed through seagrass beds at an average speed of 3.7–5.6 km h$^{-1}$ for approximately 2 min, resulting in linear distances of approximately 116.7 m $\pm$ 0.12 SE per trawl. Latitude and longitude were recorded at the beginning, midpoint, and end of each trawl to record the trawl path to obtain accurate distance calculations and provide coordinates for sled and quadrat surveys. Macroalgal abundance was determined as the wet weight of algae present in the trawl, measured using a spring scale. Samples of drift algae were taken from the trawl, bagged, frozen, and later identified to genus using a dissecting microscope (*Littler & Littler, 2000*). Due to variations in the trawl sampling distance, drift algal weight was standardized to trawl area (g m$^{-2}$). At the center of each trawl path, environmental parameters including salinity, temperature (°C), and dissolved oxygen (mg L$^{-1}$) were measured using a YSI Pro 2030 containing a galvanic DO sensor (Model 2002) immediately following the trawl. Water depth (cm) was also measured at the center of the trawl path.

We returned to each site to sample using the epibenthic sled, which consisting of an aluminum frame (0.75-m wide and 0.6-m high), with two skids on either side (0.8 m in length), and a 2-mm stretch mesh net. Sled samples were collected near the midpoint of the trawl path. The sled was towed for 13.3 m at approximately 0.5 m sec$^{-1}$, covering an area of 10 m$^2$. Algae from benthic sled samples were bagged, frozen, and transported to the lab where they were later identified to genus and weighed.

**Table 1 Abiotic variables from each region.** Abiotic variables (Mean ± SE) measured during the early (May–June) and late (August–September) summer months 2018.

| Abiotic parameter | Time | Laguna Madre, TX | Corpus Christi Bay, TX | Chandeleur Islands, LA | Cedar Key, FL | Charlotte Harbor, FL |
|---|---|---|---|---|---|---|
| Salinity (ppt) | Early | 36.8 ± 0.06 | 33.9 ± 0.16 | 16.2 ± 0.49 | 28.3 ± 0.59 | 22.6 ± 1.22 |
| | Late | 37.1 ± 0.13 | 34.9 ± 0.14 | 27.6 ± 0.27 | 25.1 ± 0.70 | 19.9 ± 0.96 |
| Temperature (°C) | Early | 28.1 ± 0.20 | 27.0 ± 0.15 | 29.8 ± 0.55 | 29.7 ± 0.16 | 30.1 ± 0.22 |
| | Late | 29.7 ± 0.38 | 30.2 ± 0.20 | 30.8 ± 0.25 | 30.1 ± 0.52 | 31.8 ± 0.29 |
| Dissolved Oxygen (mg L$^{-1}$) | Early | 8.4 ± 0.43 | 6.2 ± 0.32 | 9.5 ± 0.58 | 7.6 ± 0.30 | 6.3 ± 0.24 |
| | Late | 6.1 ± 0.47 | 11.8 ± 0.46 | 8.9 ± 0.38 | 6.8 ± 0.35 | 7.7 ± 1.50 |

### Seagrass and algal vegetative sampling (quadrats)

The structural complexity of seagrass meadows and percent cover of drift algae were assessed using a 1-m$^2$ quadrat divided into 100, 10-cm × 10-cm cells. Twelve quadrats were haphazardly thrown along each trawl: four quadrats at the beginning, middle, and end of each trawl path. The percent cover, shoot count, and canopy height of each seagrass species present in a quadrat was recorded. Seagrass percent cover by species, as well as the cover of drift algae, were measured by counting the number of grid cells within each quadrat that contained a particular vegetation type (0–100 grids quadrat$^{-1}$). The shoot count was calculated for each seagrass species by counting the number of shoots within a random quadrat grid cell. Canopy height was defined as the mean of three randomly selected canopy height measurements.

### Statistical analysis

SAS (SAS Institute, Cary, NC, USA) was used for all statistical analyses. When comparing algal biomass across the northern Gulf of Mexico, analyses were performed using General Linear Models (GLM) in SAS with region (CB, CH, CK, LA, and LM) and sampling period (early or late summer) as fixed factors. The algal biomass within the trawl and epibenthic sled samples were log-transformed to mitigate skewness and achieve normality. Comparisons of algal weight across early and late summer were then performed using procedure GENMOD with link function gamma to analyze continuous variables and $\alpha = 0.05$ was maintained in all *post hoc* testing.

To compare algal percent cover to environmental and seagrass variables within the quadrats, a multiple linear regression with backward elimination model selection was performed using procedure REG in SAS, maintaining $\alpha = 0.05$ during model selection. Variables included in this regression were shoot count, average canopy height, salinity, temperature, dissolved oxygen, and the percent cover of *T. testudinum, H. wrightii,* and *S. filiforme* from the quadrats.

## RESULTS

### Macroalgal biomass across the northern Gulf of Mexico

Drift macroalgae were present in both the early and late summer sampling times, reaching biomasses of over 50 g m$^{-2}$ in one site in Charlotte Harbor, FL (Fig. 2). Regional

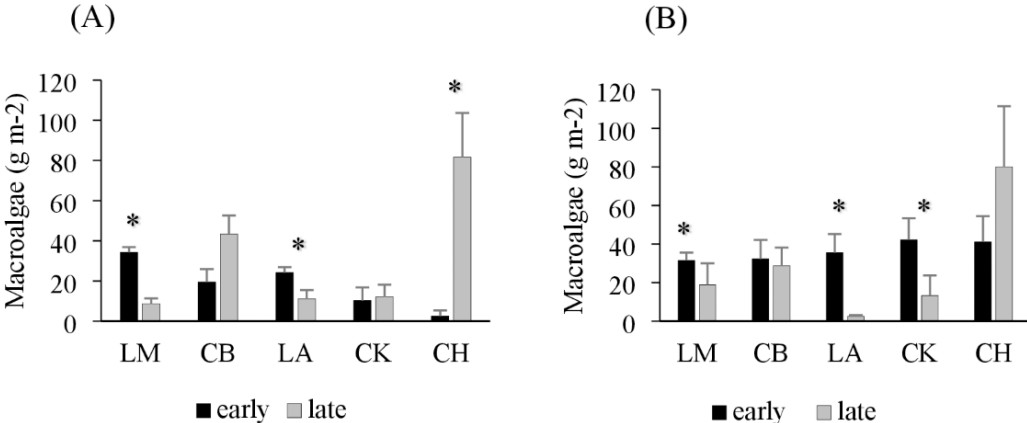

**Figure 2 Average trawl and sled macroalgal weight in each region.** (A) Average trawl macroalgae weight (g m$^{-2}$) + SE and (B) benthic sled macroalgae weight (g m$^{-2}$) + SE sampling across each region during the early (black) and late (grey) summer months. The regions include Laguna Madre, TX (LM, $n = 20$), Corpus Christi Bay, TX (CB, $n = 20$), Chandeleur Islands, LA (LA, $n = 20$), Cedar Key (CK, $n = 25$), and Charlotte Harbor, FL (CH, $n = 25$). The asterisk (*) indicates that there is a significant difference between early (May–June 2018) and late (August–September 2018) sampling within that region.

comparisons of trawl samples showed no significant changes in the overall algal biomass in Corpus Christi Bay, TX or Cedar Key, FL between early and late sampling times ($p = 0.49$ and 0.29, respectively; Table 2). Charlotte Harbor, FL had significantly higher biomass of drift macroalgae in the late summer when compared to the early summer months ($p < 0.001$; Table 2). Meanwhile, Laguna Madre, TX and Chandeleur Islands, LA both had significantly less macroalgae in the late summer months compared to the early summer ($p = 0.05$ and 0.01, respectively; Table 2). Macroalgal biomass collected in the epibenthic sled followed a similar pattern among regions and sampling times (Fig. 2, Table 2).

## Macroalgal species composition across the northern Gulf of Mexico

Drift algal community composition varied across region. Macroalgae identified in Laguna Madre, TX consisted of the genera *Agardhiella*, *Amphiroa*, *Dictyota*, *Digenia*, *Gracilaria*, *Hypnea*, and *Laurencia*. Corpus Christi Bay, TX had similar algal genera comprised of *Agardhiella*, *Chondria*, *Dictyota*, *Digenia*, *Gracilaria*, and *Laurencia*. Macroalgae found in Chandeleur Islands, LA consisted of *Agardhiella*, *Chondria*, *Gracilaria*, *Laurencia*, and *Spyridia*. Cedar Key, FL consisted of *Acanthophora*, *Dictyota*, *Digenia*, *Gracilaria*, *Laurencia*, *Polysiphonia*, and *Ulva*. Charlotte Harbor, FL were mainly comprised of *Acanthophora*, *Cladophora*, *Gracilaria*, and *Spyridia* genera, and *Hypnea*, *Laurencia*, and *Ulva* were also present.

## Macroalgal percent cover in relation to seagrass and abiotic parameters

Within the quadrat surveys, *T. testudinum* percent cover and average canopy height both significantly related to drift algal percent cover ($p = 0.03$ and $p < 0.001$, respectively; Fig. 3), whereas the seagrass shoot counts, salinity, temperature, and percent cover of *H.*

**Table 2  Comparisons of trawl and sled algal weight across region and sampling time.** Multiple and general linear regression models for macroalgal biomass across region during early (May–June 2018) and late (August–September 2018) summer sampling. Each location was then separated and analyzed individually using generalized estimating equations with sample period as the fixed factor.

| | SS | df | F ratio | Prob >F |
|---|---|---|---|---|
| *Trawl algae weight* | | | | |
| Region (LM, CB, LA, CK, CH) | 1937715.58 | 4 | 6.87 | <0.0001 |
| Sample period (early, late) | 142138.139 | 1 | 2.52 | 0.1137 |
| Region*sample period | 2547045.91 | 4 | 9.03 | <0.0001 |
| *Sled algae weight* | | | | |
| Region (LM, CB, LA, CK, CH) | 197.888 | 4 | 3.38 | 0.0104 |
| Sample period (early, late) | 92.383 | 1 | 28.99 | <0.0001 |
| Region*sample period | 65.214 | 4 | 2.39 | 0.0521 |

| Algal biomass comparisons across sampling time | Mean estimate | Chi-Square | Pr >ChiSq |
|---|---|---|---|
| *Trawl* | | | |
| LM (early v late) | −479.345 | 6.51 | 0.0107 |
| CB (early v late) | 3773.820 | 0.49 | 0.4859 |
| LA (early v late) | −527.146 | 3.86 | 0.0493 |
| CK (early v late) | −1150.98 | 1.11 | 0.2921 |
| CH (early v late) | 299.2744 | 14.22 | 0.0002 |
| *Sled* | | | |
| LM (early v late) | −26.07 | 4.97 | 0.7533 |
| CB (early v late) | 238.02 | 0.10 | 0.3665 |
| LA (early v late) | −13.46 | 18.31 | 0.2010 |
| CK (early v late) | −28.12 | 1.64 | <0.0001 |
| CH (early v late) | 88.59 | 0.82 | 0.0258 |

*wrightii* and *S. filiforme* did not significantly contribute to the drift algae cover in these areas ($p > 0.05$; Table 3). Drift algal cover increased with increasing *T. testudinum* cover, and algae were most dense in areas with intermediate seagrass canopy heights around 400 mm tall (Fig. 3).

## DISCUSSION

Drift macroalgae were found throughout the study region in both the early and late summer months, with lower biomasses observed in the late summer months across Laguna Madre, TX, Cedar Key, FL, and Chandeleur Islands, LA, consistent with previous findings (*e.g., Benz, Eiseman & Gallaher, 1979*; *Virnstein & Carbonara, 1985*). Conversely, macroalgal abundance was higher in the late summer in Charlotte Harbor, FL, and did not significantly change from early to late summer in Corpus Christi Bay, TX. The inconsistent patterns observed in these two estuaries may be the result of tidal and freshwater variations that affect delivery rates of nutrients and/or flushing of drift macroalgae. Corpus Christi Bay, TX is an enclosed system with little direct influence from the Gulf of Mexico and long water residence times (*Solis & Powell, 1999*; *Pulich Jr, 2007*). Since macroalgae are commonly flushed by tidal currents, the long water residence time and protection from
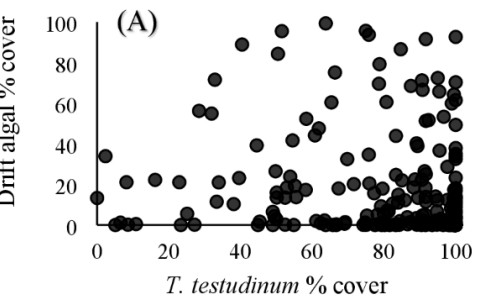
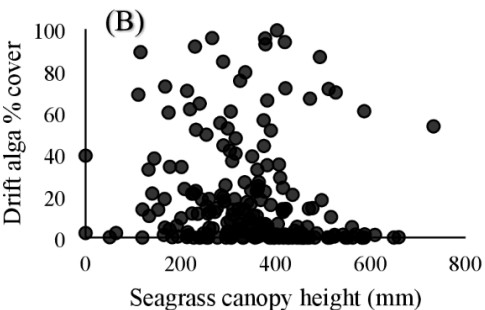

**Figure 3** **Vegetation comparisons.** (A) Scatterplots comparing the drift macroalgae percent cover to *T. testudinum* percent cover and (B) average seagrass canopy height from the quadrat dataset. (A) displays drift algae cover increasing with increased *T. testudinum* cover and (B) shows algae percent cover was highest at intermediate canopy heights (∼400 mm).

**Table 3** **Backward regression to compare algae to abiotic and seagrass characteristics.** Multiple linear regression with backward elimination model selection results comparing algal density across abiotic and seagrass variables. The variables included in the model, following selection, include *T. testudinum* density and the average canopy height. Variables that were determined to be insignificant to the model during the selection process were average shoot count, salinity, water temperature, *H. wrightii* density, *S. filiforme* density, and dissolved oxygen.

| | SS | df | F ratio | Prob >F |
|---|---|---|---|---|
| *Final model* | 4.87 | 2 | 7.10 | 0.0010 |
| *Variables in final model* | | | | |
| *T. testudinum* density | 1.73 | | 5.04 | 0.0258 |
| Average canopy height | 4.50 | | 13.11 | 0.0004 |

| | Model R² | C(p) | F ratio | Prob >F |
|---|---|---|---|---|
| *Variables removed from the model* | | | | |
| Average total shoot count | 0.0760 | 7.02 | 0.02 | 0.8964 |
| Salinity | 0.0749 | 5.26 | 0.25 | 0.6182 |
| Temperature | 0.0737 | 3.53 | 0.27 | 0.6028 |
| *H. wrightii* density | 0.0702 | 2.36 | 0.83 | 0.3626 |
| Dissolved oxygen | 0.0652 | 1.52 | 1.18 | 0.2791 |
| *S. filiforme* density | 0.0604 | 0.65 | 1.14 | 0.2862 |

tidal flushing in Corpus Christi Bay may reduce seasonal algae decline observed in other locations. The high percentage of clay and silt within the benthic sediments of Corpus Christi Bay further indicates differences in the local hydrodynamic regime (*Shideler, Stelting & McGowen, 1981*), with previous studies showing a direct positive link between algae biomass and the amount of silt-clay in the system (*Bell & Hall, 1997*). In contrast, Charlotte Harbor, FL, has experienced an increase in macroalgal blooms in recent years, particularly on the eastern shore (*BTT, 2021*). The increase in nutrient concentrations from creeks and streams, as well as the limited water circulation on the east side of Charlotte Harbor, appears to be a major driver of high accumulation of drift algae in these locations. Nutrient inputs from leaking septic systems, fertilizer and agricultural runoff, untreated

stormwater, ineffective sewage treatment systems, and altered freshwater inflow have been deemed the primary cause of these algae blooms (*Lapointe et al., 2016*; *BTT, 2021*). While seasonal fluctuations often show a decline in algal abundances in the late summer months, localized anthropogenic and hydrodynamic differences may cause localized variations.

Drift macroalgae were most dense in areas with higher percent cover of *T. testudinum* but with an intermediate canopy height (∼400 mm). Perhaps unsurprisingly, algae are more likely to be entrained within denser seagrass beds (*Virnstein & Carbonara, 1985*; *Bell & Hall, 1997*), but this varies depending on location and scale. For example, at smaller spatial scales (m), shoot count and blade length were associated with algal density patterns in Tampa Bay, FL (*Bell, Hall & Robbins, 1995*). However, when this same location was studied at a larger scale (km), seagrass cover explained 57% of the variation in algal cover, suggesting that spatial scale is important when comparing algal to seagrass communities (*Bell & Hall, 1997*). The relationship between seagrass canopy height and macroalgal density could be related to light-limiting growth restrictions. When macroalgae reach sufficient biomass, they can restrict the light available to seagrasses, decreasing productivity (*Hauxwell et al., 2001*; *Huntington & Boyer, 2008*). This may be why areas with the tallest seagrass canopies also have less macroalgae. Because light is less of a limiting factor, seagrass communities can grow at a faster rate when macroalgae are not present. Conversely, entrapment of algae at intermediate canopy heights could be related to the interaction between the algae and flow conditions. Taller canopies, that extend closer to the surface of the water may be influenced by higher water velocities and turbulence, increasing the likelihood of macroalgae dislodgement.

This study was limited in its ability to determine species-specific algal effects across varying seagrass characteristics. During sampling, macroalgal species were not separated, identified, and weighed in the field, but were rather weighed collectively and a sample of each species brought back to the lab for later identification. Preliminary site selection was also chosen based on the presence of *T. testudinum* across all site locations, which could have obscured less obvious patterns when comparing algae biomass to *S. filiforme* and *H. wrightii*. Another potential reason for the seagrass species-specific differences are the varying morphological characteristics across *T. testudinum*, *S. filiforme*, and *H. wrightii*, including differences in blade width, thickness, and shoot height (*Loria, 2019*). Future research should further investigate concentrations of drift macroalgae in *S. filiforme* and *H. wrightii* dominated seagrass meadows to corroborate our nonsignificant findings.

Macroalgae proliferate when sufficient light, nutrient availability, and warm temperatures co-occur (*EPA, 2013*) and can become entrained in seagrass beds in large quantities when seagrasses are dense and water flow is low enough to prevent dislodgement (*Bell & Hall, 1997*). Although algae can be beneficial by increasing habitat complexity and food resources within seagrass meadows (*e.g.*, *Carr, 1994*; *Kingsford, 1995*; *Jones, Lawton & Shachak, 1997*; *Morris & Hall, 2001*; *Guido et al., 2004*), large blooms of macroalgae can displace other benthic habitats and reduce the health of the seagrass meadows (*Valiela et al., 1997*; *Tagliapietra et al., 1998*). This study identified several patterns in algal density and composition on a Gulf-wide scale. Data collected during this study also found large abundances of small fish and invertebrates living within these seagrass beds, indicating that

the biomass of algae present in the northern Gulf of Mexico do not appear to be detrimental (*Belgrad et al., 2021*; *Correia, 2021*). Given the extent that drift algae spatiotemporally vary, research to quantify links between algal genera and faunal community composition could resolve much of the uncertainty surrounding this relationship. As the climate continues to change, macroalgal blooms may become more variable and understanding the interaction between algal-fauna relationships becomes increasingly important. Understanding the role of macroalgae within Gulf of Mexico seagrass beds will allow us to better manage the fisheries and other coastal resources in the future.

## ACKNOWLEDGEMENTS

The authors would like to thank Samantha Smith for assistance in the field and laboratory, Manuel Merello for identifying macroalgal samples, and the reviewers who provided helpful edits to the manuscript prior to publication.

### Funding

This paper is a result of research funded by the National Oceanic and Atmospheric Administration's RESTORE Science award # NA17NOS4510093 to the University of Southern Mississippi, Dauphin Island Sea Lab, University of Florida, and Florida Fish and Wildlife Conservation Commission. The funders had no role in study design, data collection and analysis, decision to publish, or preparation of the manuscript.

### Grant Disclosures

The following grant information was disclosed by the authors:
National Oceanic and Atmospheric Administration's RESTORE Science: # NA17NOS4510093.

### Competing Interests

The authors declare there are no competing interests.

### Author Contributions

- Kelly M. Correia performed the experiments, analyzed the data, prepared figures and/or tables, authored or reviewed drafts of the article, and approved the final draft.
- Scott B. Alford performed the experiments, authored or reviewed drafts of the article, and approved the final draft.
- Benjamin A. Belgrad performed the experiments, authored or reviewed drafts of the article, and approved the final draft.
- Kelly M. Darnell conceived and designed the experiments, performed the experiments, authored or reviewed drafts of the article, and approved the final draft.
- M. Zachary Darnell conceived and designed the experiments, performed the experiments, authored or reviewed drafts of the article, and approved the final draft.
- Bradley T. Furman performed the experiments, authored or reviewed drafts of the article, and approved the final draft.

- Margaret O. Hall performed the experiments, authored or reviewed drafts of the article, and approved the final draft.
- Christian T. Hayes performed the experiments, authored or reviewed drafts of the article, and approved the final draft.
- Charles W. Martin performed the experiments, authored or reviewed drafts of the article, and approved the final draft.
- Ashley M. McDonald performed the experiments, authored or reviewed drafts of the article, and approved the final draft.
- Delbert L. Smee performed the experiments, authored or reviewed drafts of the article, and approved the final draft.

## Data Availability

   Data used in statistical analyses are available as a Supplemental File.

## Supplemental Information

Supplemental information for this article can be found online at http://dx.doi.org/10.7717/peerj.13855#supplemental-information.

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
