# Peer review of "Drift macroalgal distribution in northern Gulf of Mexico seagrass meadows"

_PeerJ, doi:10.7717/peerj.13855_

## Round 0.1 · original submission · Major Revisions

Please consider the comments of the reviewer carefully!

Reviewer 1 ·

Basic reporting

No comment - article meets all expectations

Experimental design

No comment - article meets all expectations

Validity of the findings

No comment - article meets all expectations

Additional comments

See minor suggestions in the attached PDF

Annotated reviews are not available for download in order to protect the identity of reviewers who chose to remain anonymous.

Reviewer 2 ·

Basic reporting

The authors present a well-written, concise manuscript on drift algal cover and biomass at sites across the Gulf Mexico. The article is structured well, with sufficient figures and tables. I think the manuscript can benefit on a little more information about drift algae - particularly as an ephemeral feature as well as a potentially important vector for dispersal of small marine organisms. I think the ephemeral aspect can be particularly important to put into context of other literature.

Experimental design

The design seems appropriate to address their research questions. The authors used multiple techniques to estimate algal abundance and biomass - which did differ slightly in the outcomes. One thing that I think would benefit the manuscript, however, is algal species-level or genus-level analysis. There is a lot of variability in the relationships with Thalassia, and I am curious if some of that variability could be explained by not lumping all drift algae together but by separating out different groups. It also may be my naivete about some of these macroalgae, but are all of them considered drift algae? For example, in my experience, Dictyota grows attached to the substrate, and so I would not consider it a drift macro, and so when you catch it in trawls may be more reflective of the substrate environment in the different regions. Otherwise, the methods are described in sufficient detail and can be replicated.

Validity of the findings

I think the discussion could use some restructuring. Given the spot measurement nature of this study, and differences of methods between this and earlier studies, it is really difficult to determine how useful any of the comparisons of biomass are and what it trying to be conveyed with them. It was never mentioned in the introduction or objectives that this would be used to compare across time, so it is not clear why that seems to be a big part of the discussion. I think the discussion would benefit from restructuring focusing on the primary objectives stated in the introduction: 1) establish the baseline of drift algae across the northern GOM in the summer and 2) explore the relationship with seagrass metrics. I think it would be useful in talking about the trends in algal biomass across the locations to also include something about the differences in species composition and diversity. Given differences in sampling methods, that might be something more useful to track changes over time. In addition, I think the discussion would benefit in some explanations as to why the other two common seagrass species did not show any relationships with algae (morphology I would assume?) and also what it means for there to be more algae with increasing seagrass density (when, as mentioned in the intro and discussion, macroalgae is bad for seagrasses).

---

## Round 0.2 · Major Revisions

Please consider the reviewer's comments carefully.

Reviewer 2 ·

Basic reporting

The manuscript is well-written and has included numerous changes based on reviewer suggestions. It is structured well, with sufficient tables and figures. I do have concerns that the authors are still trying to make the manuscript 'bigger than it is' despite reviewer feedback by trying to go beyond the objectives stated in the manuscript.

"Consequently, the functional role of drift algae may increase
84 in importance in regions where seagrasses have declined. However, drift algae are known for
85 being highly variable across space and time (Benz, Eiseman & Gallaher, 1979; Bell & Hall,
86 1997), with their abundance and movement within estuaries rarely quantified and difficult to
87 track, complicating our understanding of drift algae in seagrass ecosystems. "

To me, this is the main point of doing it - this is an observational study across a wide geographic scope, and its ok that this is purely observational. By trying to make comparisons after the fact, despite vastly different methods between this and earlier studies (will point out more problems below). This study has value. As it is framed, the value is being lost.

Experimental design

It's unfortunate that the authors cannot examine taxon-level relationships with the drift algae and seagrass and unfortunately, detracts from the overall manuscript and it's value. It also creates major concerns about wet wt to dry wt - each algal species is going to have a different relationship and just lumping all algae together and saying this is the equivalent dry weight is not valid. This was not clear the first time through the manuscript and makes it challenging to do any of the comparisons the authors are trying to do. If the data exists within Cedar Key, that can certainly be reported - 20 sites and 2 timepoints = 40 data points for taxon=specific seagrass metric relationships. I know its not great, and it is not at all sites, but it can be very useful.

Validity of the findings

I am just going to repeat what I said last time: given the spot measurement nature of this study AND dramatic differences in methods between this (trawling) and other studies (quadrat counts, transects), it is difficult to determine how useful the biomass comparisons are. Further, despite the authors assertions, it is not even valuable for percent occurrence - the trawls are sampling a much larger area than quadrat samples or transect surveys! Of course they are going to find algae more frequently! The only thing that might be valuable is species level changes - as different macroalgal species could be useful as indicator species. But its not clear the authors can do that. Furthermore, many of the comparisons aren't even relevant. For example, what happens in Charlotte Harbor is not comparable to Florida Bay - a VASTLY different system, nor the IRL which is on the Atlantic coast and has, again, a vastly different morphology, flushing regime, and land use. This is still the major part of the discussion, and it really seems like authors trying to make a paper "bigger" than it is. Again, there is nothing wrong with this being an observational paper just describing the macroalgal abundance during peak times at different locations spanning the GOM, particularly if we expect macroalgae to increase and replace seagrass. That is fine.
So, as I said last time, the discussion would benefit from restructuring focusing on the primary objectives stated in the introduction: 1) establish the baseline of drift algae across the northern GOM in the summer and 2) explore the relationship with seagrass metrics.

---

## Round 0.3 · accepted · Accept

I congratulate the authors for the effort put into this paper! The manuscript is significantly improved; therefore, I recommend accepting it in its current form!

Reviewer 2 ·

Basic reporting

The authors do a nice job with providing a well-written, succinct manuscript that focuses solely on the initial questions and the work that was done.

Experimental design

The experimental design is valid.

Validity of the findings

I appreciate this new version of the manuscript that focuses specifically on what was done and what can be gleaned from that information without trying to make the implications too broad. Nice job!